# Gut Microbiota Profile Changes in Patients with Inflammatory Bowel Disease and Non-Alcoholic Fatty Liver Disease: A Metagenomic Study

**DOI:** 10.3390/ijms25105453

**Published:** 2024-05-17

**Authors:** Carmen De Caro, Rocco Spagnuolo, Angela Quirino, Elisa Mazza, Federico Carrabetta, Samantha Maurotti, Cristina Cosco, Francesco Bennardo, Roberta Roberti, Emilio Russo, Amerigo Giudice, Arturo Pujia, Patrizia Doldo, Giovanni Matera, Nadia Marascio

**Affiliations:** 1Health Sciences Department, University “Magna Graecia” of Catanzaro, 88100 Catanzaro, Italy; decaro@unicz.it (C.D.C.); quirino@unicz.it (A.Q.); federico.carrabetta@studenti.unicz.it (F.C.); francesco.bennardo@unicz.it (F.B.); roberta.roberti9@outlook.com (R.R.); erusso@unicz.it (E.R.); a.giudice@unicz.it (A.G.); mmatera@unicz.it (G.M.); nmarascio@unicz.it (N.M.); 2Experimental and Clinical Medicine Department, University “Magna Graecia” of Catanzaro, 88100 Catanzaro, Italy; elisamazza@unicz.it (E.M.); smaurotti@unicz.it (S.M.); doldo@unicz.it (P.D.); 3Unit of Gastroenterology and Operative Endoscopy, University Hospital “Renato Dulbecco” of Catanzaro, 88100 Catanzaro, Italy; cosco.cristina@libero.it; 4Medical and Surgical Sciences Department, University “Magna Graecia” of Catanzaro, 88100 Catanzaro, Italy; pujia@unicz.it

**Keywords:** gut microbiota, IBD, NAFLD, liver steatosis, leaky gut

## Abstract

Gut microbiota imbalances have a significant role in the pathogenesis of Inflammatory Bowel Disease (IBD) and Non-Alcoholic Fatty Liver Disease (NAFLD). Herein, we compared gut microbial composition in patients diagnosed with either IBD or NAFLD or a combination of both. Seventy-four participants were stratified into four groups: IBD-NAFLD, IBD-only, NAFLD-only patients, and healthy controls (CTRLs). The 16S rRNA was sequenced by Next-Generation Sequencing. Bioinformatics and statistical analysis were performed. Bacterial α-diversity showed a significant lower value when the IBD-only group was compared to the other groups and particularly against the IBD-NAFLD group. β-diversity also showed a significant difference among groups. The higher Bacteroidetes/Firmicutes ratio was found only when comparing IBD groups and CTRLs. Comparing the IBD-only group with the IBD-NAFLD group, a decrease in differential abundance of Subdoligranulum, Parabacteroides, and Fusicatenibacter was found. Comparing the NAFLD-only with the IBD-NAFLD groups, there was a higher abundance of Alistipes, Odoribacter, Sutterella, and Lachnospira. An inverse relationship in the comparison between the IBD-only group and the other groups was shown. For the first time, the singularity of the gut microbial composition in IBD and NAFLD patients has been shown, implying a potential microbial signature mainly influenced by gut inflammation.

## 1. Introduction

The gut microbiota refers to the microbial community that inhabits the gastrointestinal tract and consists of various microorganisms, with bacteria being the predominant species. *Firmicutes* and *Bacteroidetes* make up nearly 90% of all phyla [1]. The intestinal microbiota plays a crucial role in maintaining the body’s homeostasis, supporting the proper functioning of several physiological systems, ranging from the immune system to the brain. Consequently, it is also implicated in the development of various diseases [2,3]. Inflammatory Bowel Diseases (IBDs), including Crohn’s Disease (CD) and Ulcerative Colitis (UC), are chronic inflammatory disorders predominantly affecting the gastrointestinal tract, characterized by an abnormal immune response induced by environmental stimuli in genetically predisposed subjects [4]. One of the extensively researched changes associated with the pathogenesis of IBD is the abnormal permeability of the gut barrier, commonly referred to as “leaky gut”, which leads to an induction of the inflammatory response through the exposure of immune cells to various microbial-associated substances. The dysfunction of the gut barrier is influenced by genetic factors, environmental elements, and gut dysbiosis [5]. Studying the active role of the gut microbiota through an integrated multi-omics approach provides numerous tools to gain biomarkers useful for both diagnostic and prognostic purposes [6]. Furthermore, gut microbiota is also implicated in numerous extraintestinal manifestations of IBD, including various metabolic comorbidities [7].

Non-Alcoholic Fatty Liver Disease (NAFLD) is a comprehensive term that includes the accumulation of fat in the liver (>5% of hepatocytes) and non-alcoholic steatohepatitis (NASH), which can progress to liver cirrhosis and hepatocarcinoma [8]. NAFLD has a multifaceted pathogenesis, encompassing genetic and environmental factors, dietary habits, and metabolic disorders such as insulin resistance. These factors contribute, through lipotoxicity and other mechanisms such as dysbiosis, to the disruption of immune homeostasis, which is responsible for the overlapping of several dysmetabolic conditions within the context of metabolic-dysfunction-associated fatty liver disease (MAFLD) [9].

Metabolic-Dysfunction-Associated Steatotic Liver Disease (MASLD) is a more inclusive and non-stigmatizing term for NAFLD proposed by experts, which refers to the simultaneous evidence of hepatic steatosis and at least one of five cardiac metabolic risk factors [10].

The prevalence of NAFLD in the general population has increased significantly over time, from 25.5% before 2005 to 37.8% in 2016 or later [11]. A recent epidemiologic study investigating the occurrence of hepatobiliary conditions in IBD patients revealed a median 26% prevalence of NAFLD [12,13]. Crucially, patients with IBD exhibit an elevated rate of NAFLD diagnosis independent of traditional metabolic risk factors, such as obesity and diabetes [14].

To elucidate the pathogenetic interplay between IBD and NAFLD, numerous studies have delved into both genetic and epigenetic factors, as well as gut dysbiosis. Intriguingly, a previous study has demonstrated how carriers of the PNPLA3 148M allele mutation with IBD have a greater risk of hepatic steatosis and increased circulating alanine transaminase (ALT) [15]. Alterations in the relative abundance of microbial species have been documented in various studies examining the intestinal microbial composition in patients with IBD or NAFLD. These studies have reported a decrease in strains that produce short-chain fatty acids (SCFA) and an increase in strains that degrade colonic mucin, even though the alterations of the microbiota in the presence of both disorders are not entirely clear and have been poorly elucidated [16]. Moreover, the role of medical treatments, including glucocorticoids, immunomodulators, and tumor necrosis factor-α inhibitors, in the pathogenesis of NAFLD in IBD also remains unclear [17]. Alongside the diagnostic role of the microbiota in the context of these diseases, there is a growing interest in microbiome-targeted therapies for their management, such as probiotics, synbiotics, antibiotics, and fecal microbiota transplantation (FMT) [18,19]. Metagenomic approaches enable us to preliminarily highlight how dysbiosis may be involved in both gut and liver disorders.

The aim of this study was to evaluate the composition of the gut microbiota in patients with both IBD and NAFLD, comparing the results with each disorder considered separately.

## 2. Results

### 2.1. Patients’ Characteristics

Seventy-four participants (56 patients and 18 healthy controls) passed the dental examination according to the defined criteria and were included in the study. The participants were stratified into four groups according to their IBD diagnosis and the presence of NAFLD measured by a Controlled Attenuation Parameter (CAP). Anthropometric and clinical features of the four cohorts are summarized in Table 1. The first group, named IBD-NAFLD, included 18 IBD patients; 78% (*n* = 14) were male with a median age of 45 years (IQR = 37–53). The second group, named IBD-only group, included 20 IBD patients; 55% (*n* = 11) were male with a median age of 39 years (IQR = 24–45). The third group, named NAFLD-only group, included 18 patients; 72% (*n* = 13) were male with a median age of 42 years (IQR = 36–56). The fourth group included 18 healthy controls (CTRLs); 61% (*n* = 11) were male with a median age of 33 years (IQR = 23–49). As expected, median Body Mass Index (BMI) values were found to be significantly higher (*p* = 0.001) in NAFLD patients versus the other study groups (28 kg/m^2^ in the IBD-NAFLD and NAFLD-only groups vs. 24 kg/m^2^ in the IBD-only and CTRLs groups, respectively). With respect to the two IBD groups stratified by the presence or absence of NAFLD, no major differences were found according to disease characteristics, as shown in Table 2. In the IBD-NAFLD group, 67% (*n* = 12) of patients had UC, and 33% (*n* = 6) had CD, while in the IBD-only group, 50% (*n* = 10) of patients had UC, and the other half (*n* = 10) had CD. In both groups, most CD patients had an ileocolic localization (66% in the IBD-NAFLD group, 80% in the IBD-only group) and a non-perforating–non-stricturing phenotype (100% in the IBD-NAFLD group, 90% in the IBD-only group). Most UC patients had pancolitis (83% in the IBD-NAFLD group, 70% in the IBD-only group). Median disease duration was significantly higher in the IBD-NAFLD group than in the IBD-only group [13 years (IQR = 5–17) vs. 5 years (IQR = 2–11), respectively; *p* = 0.01]. Both cohorts were in clinical remission according to the Harvey–Bradshaw Index (HBI) and Mayo Score (MS). Indeed, no major difference was observed between the IBD-NAFLD group and IBD-only group according to median MS [0.5 (IQR = 0–1) vs. 1 (IQR = 0–2), respectively; *p* = 0.49] and median HBI [6.5 (IQR = 3–7) vs. 4.5 (IQR = 0–9), respectively; *p* = 0.9]. Almost all patients were treated with mesalamine, and only 8% (*n* = 3) were on steroids. Seven patients (39%) in the IBD-NAFLD group and 11 patients (55%) in the IBD-only group were using biologic drugs, most of which were TNF-alpha inhibitors (i.e., infliximab, adalimumab, golimumab), while only two (10%) were on vedolizumab, an anti-integrin drug. Given the stratification by presence of NAFLD in the four groups, as expected, ALT values were significantly higher in the IBD-NAFLD group and in the NAFLD-only group [32 UI/L (IQR = 23–46) and 33 UI/L (IQR = 24–44), respectively] versus the IBD-only group and the CTRLs group [15.5 UI/L (IQR = 10–19) and 15.5 UI/L (IQR = 11–20), respectively] (*p* = 0.001). None of the patients had Non-Alcoholic Steatohepatitis (NASH). Median Gamma-glutamyl transpeptidase (GGT) was significantly higher in the IBD-NAFLD group than in the IBD-only group (32 UI/L and 17 UI/L, respectively; *p* = 0.002). Median LDL was significantly higher in the IBD-only group than in the NAFLD-only group (101 mg/dL and 129 mg/dL, respectively; *p* < 0.05). Fibrosis was not found in any patient. Median Stiffness was 5 kPa (IQR = 4–6) for the IBD-NAFLD group, 5.3 kPa (IQR = 4–5) for the IBD-only group, 4.7 kPa (IQR = 3–5) for the NAFLD-only group and 4.4 kPa (IQR = 4–6) for the CTRLs group.

### 2.2. Gut Microbiota Analysis and Comparisons

The four study groups showed differences in the α- and β-diversity assessed by Shannon Index and PCoA, respectively (Figure 1A,B). As described below, each group was compared with the others, drawing attention to the specific differences in the abundance of OTUs at genus and species levels.

#### 2.2.1. Characteristics of Microbiota Composition in IBD-Only Group

The α-diversity analysis showed that there was a significant (*p* = 0.011) difference in the Shannon Index between the CTRLs and IBD-only groups (Figure 2A), whereas no significant difference was identified in β-diversity. At the phylum level, *Bacteroidetes*/*Firmicutes* ratio was significantly higher in the IBD-only group compared to the CTRLs group (*p* = 0.044) (Figure 2B). Different taxonomy abundance (LEfSe: LDA score > 2.0) was found in IBD patients at genus and species levels, as shown in Figure 2C. At the genus level, a reduction in *Alistipes*, *Parabacteroides*, *Subdoligranulum*, and *Fusicatenibacter* abundance was revealed in the IBD-only group compared to the CTRLs group, whereas *Anaerostipes* and *Incertae_sedis* were more abundant in the IBD-only group (*p* < 0.05) (Figure 2C).

#### 2.2.2. Characteristics of Microbiota Composition in NAFLD-Only Group

Comparing NAFLD-only patients with the CTRLs group, a nonsignificant difference in α- and β-diversity was found. The *Bacteroidetes*/*Firmicutes* ratio was significantly (*p* = 0.022) lower in the NAFLD-only group (Figure 2D). At the genus level (LEfSe: LDA score > 2.0), *Subdoligranulum*, *Roboutsia*, *Sutterella*, and *Bifidobacterium* were more abundant in the NAFLD-only group (Figure 2E).

#### 2.2.3. Specific Microbiota Signature in the IBD-NAFLD Group

No differences in α- and β- diversity were found between the IBD-NAFLD group and CTRLs group. No difference in the *Bacteroidetes*/*Firmicutes* ratio was observed. At the genus level, *Alistipes*, *Odoribacter*, and *Monoglobulus* were less abundant in the IBD-NAFLD group, while *Incertae_sedis*, *Streptococcus*, *Bifidobacterium*, *Dorea*, *Romboutsia*, and *Anaerostipes* were predominant in comparison with the CTRLs group (Figure 3).

#### 2.2.4. Differences in Microbiota Composition between IBD-Only and NAFLD-Only Groups

These two groups significantly differed both for α- and β-diversities (Figure 4A,B). The predominant phyla in both groups were *Firmicutes* and *Bacteroidetes* with a small nonsignificant higher level of *Bacteroidetes* in the IBD-only group (Figure 4C). In particular, four phyla, namely, *Firmicutes*, *Patescibacteria*, *Proteobacteria* were significantly more abundant, while *Verrucomicrobia* were significantly (*p* = 0.0076) prevalent in the NAFLD-only group (Figure 4D). At the species level, according to LEfSe LDA (score > 2.0), *Alistipes shahii*, *Alistipes obesi*, *Bacteroides massiliensis*, *Parabacteroides merdae*, *Victivallales bacterium*, *Eubacterium sireum*, *Bacteroides eggerthii*, *Odoribacter splanchnicus*, *Parabacteroides goldsteinii*, were significantly more abundant (*p* < 0.05) in the NAFLD-only group (Figure 4E). On the other hand, *Bacteroides vulgatus* was significantly more abundant (*p* = 0.024) in the IBD-only group (Figure 4E).

#### 2.2.5. Differences in Microbiota Composition between NAFLD-Only and IBD-NAFLD Groups

There were no significant differences in the α- and β-diversities between these two groups. At the phylum level, the *Bacteroidetes*/*Firmicutes* ratio was significantly (*p* = 0.010) lower in the NAFLD-only group compared to the IBD-NAFLD group (Figure 5A).

According to LEfSe LDA (score > 2.0), at the genus level, *Alistipes* (*p* = 0.033), *Odoribacter* (*p* = 0.011), *Sutterella* (*p* = 0.009), and *Lachnospira* (*p* = 0.049) were significantly more abundant in NAFLD-only patients versus IBD-NAFLD patients, while *Flavonifractor* (*p* = 0.016), *Incertae_Sedis* (*p* = 0.025), *Dorea* (*p* = 0.042), and *Anaerostipes* (*p* = 0.048) were significantly less abundant in the NAFLD-only group (Figure 5B).

#### 2.2.6. Differences in Microbiota Composition between IBD-Only and IBD-NAFLD Group

The α-diversity showed a significant difference (*p* < 0.05) in the Shannon Index between the IBD-only and IBD-NAFLD groups (Figure 5C). At the phylum level, the *Bacteroidetes*/*Firmicutes* ratio was significantly (*p* < 0.0013) higher in the IBD-only group (Figure 5D). According to LEfSe LDA (score > 2.0), at the genus level, *Agathobacter* (*p* = 0.035), *Parabacteroides* (*p* = 0.003), *Subdoligranulum* (*p* = 0.015), and *Fusicatenibacter* (*p* = 0.014) were significantly less abundant in IBD-only patients. On the other hand, *Oscillibacter* were not significantly more abundant (*p* = 0.084) in IBD-only patients (Figure 5E).

## 3. Discussion

Understanding the role of the gut microbiota in the pathogenesis of IBD and NAFLD is still an area of active research. Several studies have highlighted fundamental mechanisms associated with gut microbial composition that seem to be altered in both diseases [7]. Metagenomic analyses of the gut microbiota aim to assess how microbial composition changes in terms of richness and differential abundance of predominant bacterial species [20].

In our study, for the first time, we compared metagenomic results obtained from three different groups of patients affected by IBD and NAFLD, IBD only, NAFLD only, respectively, and a control group (CTRLs).

Consistent with previous studies, we noted a decrease in richness assessed by α-diversity in IBD patients compared to controls (*p* = 0.011) and NAFLD patients (*p* < 0.05). Interestingly, in the IBD-NAFLD group, this evidence was maintained only when comparing with the IBD-only group, suggesting that while NAFLD has a somewhat neutral impact on richness, IBD imposes or is accompanied by a significant modification. This variation could differ depending on whether it is accompanied by NAFLD.

Likewise, the results expressed in terms of β-diversity support the evidence that IBD alone induces a more pronounced alteration in the composition of the microbiota. As a direct consequence, changes in the *Bacteroidetes*/*Firmicutes* ratio were found when comparing groups that had already shown reductions in α- and β-diversity. This peculiar finding would suggest that patients with just IBD have a greater imbalance in gut microbial composition than the cohort also affected by NAFLD.

The analysis of taxonomic abundances at the genus level seems to confirm the role that some species are already known to have in the pathogenetic mechanisms implicated in IBD. Lloyd-Price et al., in their multi-omics study, discovered a markedly reduced unclassified *Subdoligranulum* species in IBD, linked to a broad spectrum of metabolites such as bile acids and polyunsaturated fatty acids (PUFAs) [21].

Another study reported that a loss of *Parabacteroides goldsteinii* may exacerbate colitis in model mice [22]. Takeshita et al. showed that a decreased number of *Fusicatenibacter saccharivorans*, which is a short-chain fatty acid (SCFA) producer, correlates with UC activity and contributes to dysbiosis [23]. SCFAs are essential to the efficiency of the intestinal barrier, so a reduction in their production promotes the development of an abnormal permeability that contributes to the propagation of damage through the gut–liver axis [24]. Furthermore, Zhang et al. found that *Anaerostipes hadrus BPB5*, although increasing butyrate content in the gut, significantly aggravated colitis in dextran sulphate sodium (DSS)-treated mice while exerting no detrimental effect in healthy mice [25]. In our study, the aforementioned species showed this specific behavior only when intestinal disease prevails over NAFLD.

These data represent a novelty compared to what has been reported in studies that confront the gut microbiota of patients with NAFLD versus only the healthy population. Indeed, also at the genus level, we observe distinctive findings in the comparison of the mixed group (IBD-NAFLD) with the groups of the two separate disorders. Briefly, the microbial composition of the NAFLD-only group is closer to healthy controls when compared to the IBD-NAFLD group, unlike the IBD-only group. A reduction in the genus *Alistipes*, for example, has been linked both to the progression of liver fibrosis in NAFLD patients versus the healthy population and to the increased activity of colitis in mice [26]. A reduction in the abundance of *Dorea* has been found in non-obese patients with NASH [27], and this genus has been linked to a microbiota-dependent protective phenotype against the development of NASH through enhanced mitochondrial activity in an experimental model [28].

In our cohorts, none of the subjects had liver fibrosis, and none of the IBD patients had active disease according to the clinical and endoscopic score, as reported in Table 2.

We overcame some limitations found in similar studies, such as the impact of diet on the gut microenvironment, by selecting a population characterized by similar eating habits according to the food frequency questionnaire [29]. The subjects in our study were also screened for periodontal disease, so as to rule out the possible influence of oral dysbiosis on the gut microbial composition [30].

Comparing the IBD patients’ group and taking into account the role of an active inflammatory environment, the microbial composition of the gut did not show significant differences when stratified according to inflammatory indices (fecal calprotectin and C-reactive protein) and the use of specific medications (TNF-alpha inhibitors and anti-integrin drugs). This aspect, although it poses a limitation in our ability to compare our data with the existing literature, also prompts us to consider a potential question: whether the combined impact of these two conditions could affect the microbial composition of the gut in a mutually influential manner. Other than that, the main limitations of our study are the small sample size, which comprises only patients in remission, and the lack of a longitudinal design that can follow changes in the microbiota as the disease’s natural history progresses. Further investigations on a larger population including patients with active disease are needed, also considering the wide range of available therapies and the pathogenetic implications of some new biomarkers, such as miRNAs [31]. Although our results do not show significant differences related to the various treatments, future well-designed longitudinal omics studies could assess the role of dysbiosis in therapeutic management and in the development of innovative tailored approaches [32].

Moreover, it is important to take into account the newly defined criteria for diagnosing MASLD, aiming to address the heterogeneity of fatty liver diseases and enforce the search for more accurate biomarkers [33]. The purely taxonomic information obtained from our study needs to be confirmed by functional studies on the actual role that the species found to be most prevalent play in the inflammatory microenvironment typical of these diseases. Encouraging results are beginning to emerge from metabolomics and metaproteomics studies, which are a key tool for confirming the pathogenetic significance of dysbiosis in IBD and NAFLD [34].

## 4. Materials and Methods

### 4.1. Ethics Statement

The present study was approved by the Ethical Committee of Calabria Region (#150, 22 April 2021) and was conducted in accordance with the Declaration of Helsinki (64th WMA General Assembly, Fortaleza, Brazil, October 2013). The patients’ clinical data were treated in agreement with the principles of good clinical practice.

### 4.2. Study Cohort Recruitment

Consecutive IBD outpatients aged between 18 and 80 years were enrolled at the University Hospital “Renato Dulbecco” (Catanzaro, Italy) between April and July 2021. Subjects diagnosed either with Crohn’s Disease or with Ulcerative Colitis based on clinical, endoscopic, and histopathological features were included. Patients underwent hepatic steatosis evaluation (see Section 4.6) and were divided into two different cohorts according to NAFLD presence. Moreover, as an appropriate control population, a cohort of consecutive NAFLD patients and healthy subjects such as patients’ family members and caregivers were also enrolled in the study. Patients and healthy controls (CTRLs) were matched by age, sex, and diet pattern measured by a frequency food questionnaire.

Exclusion criteria for each group were pregnancy or breastfeeding, reported chronic liver diseases (including viral hepatitis) or cardiovascular diseases, a Diabetes Mellitus (DM) diagnosis, periodontitis, previous abdominal surgery, antibiotic therapy (in the last 3 months), supplementation with probiotics or prebiotics in the previous 4 weeks, and PPI treatment in the previous 10 days.

### 4.3. Clinical Evaluation and Metabolic Assessment

Anthropometric data were collected from all participants by the Unit of Clinical Nutrition, University Hospital “Renato Dulbecco” of Catanzaro, and clinical parameters such as Body Mass Index (BMI) were calculated. Information about smoking status, alcohol consumption, concomitant presence of classic cardiovascular risk factors, and the use of potential hepatotoxic medications was collected during clinical interviews. All participants provided a blood sample to evaluate the following laboratory parameters: triglycerides, total cholesterol, low-density lipoprotein (LDL) cholesterol, high-density lipoprotein (HDL) cholesterol, glucose, alanine aminotransferase (ALT), aspartate aminotransferase (AST), gamma glutamyl transferase (GGT), total and fractionated bilirubin, C-reactive protein (CRP), and Erythrocyte Sedimentation Rate (ESR). We obtained a stool sample to evaluate fecal calprotectin through a chemiluminescence immunoassay (CLIA), using 100 mcg/g as the threshold to define disease activity.

### 4.4. Oral Examination

On the same day that the samples were taken, a dental visit was performed at the Dental School of the Magna Graecia University of Catanzaro. An experienced dentist performed the clinical evaluation of the patients’ oral health status. The oral examination was performed in accordance with the WHO-standardized methodology [35]. The decayed, missing, and filled teeth (DMFT) index was used to record the sum of decayed, missing, or filled dental elements. Periodontal health was also evaluated using two scoring systems (Gingival Index, GI; Plaque Index, PI; Table 3) as previously reported by Bianco et al. [36]. Similarly, GI and PI are scored for six index teeth. If any of these six teeth were missing, one of the adjacent teeth was examined. The dentist also recorded whether the patient was edentulous or wore dentures. GI and PI were not evaluated in fully edentulous patients. High scores indicate worse dental health. Patients with more than N decayed teeth or with GI mean > 2.5 or with GI mean > 2 and PI mean > 2.5 were excluded from the study.

### 4.5. IBD Patients’ Assessment

Patients with a diagnosis of Inflammatory Bowel Disease were recruited from the Unit of Gastroenterology, University Hospital “Renato Dulbecco” of Catanzaro. Information about disease duration, disease extension (ileal, ileocolonic, isolated upper gastrointestinal locations for CD or proctitis, left-side colitis, pancolitis for UC), disease phenotype (non-stricturing–non-perforating, stricturing, perforating for CD), and complications was collected. Disease activity was evaluated according to clinical and laboratory parameters. A Harvey–Bradshaw Index above 7 points for CD [37] and Mayo Score above 5 points for UC [38], along with serum CRP, ESR, and fecal calprotectin levels, were indicative of active disease. Information about previous and current treatment (Mesalamine, Steroids, Thyopurine, TNF-alpha inhibitors, and other biologic drugs) was also collected.

### 4.6. NAFLD Diagnosis

All participants underwent Transient Elastography (FibroScan®, Echosens, Paris, France) to evaluate the presence of hepatic steatosis [39]. Controlled Attenuation Parameter (CAP; Echosens, Paris, France) measured by TE was used to diagnose and quantify the degree of hepatic steatosis. The cutoff used to determine the presence of steatosis was 248 dB/m for >S0.

### 4.7. Next-Generation Sequencing (NGS) of Gut Microbiota

Bacterial genomic DNA was extracted from fecal samples stored in OMNIgene^®^•GUT, using the QIAamp DNA Stool Mini Kit according to manufacturer’s instructions (Qiagen, Hilden, Germany). DNA concentration was measured fluorometrically using a Qubit dsDNA BR assay kit (ThermoFisher Scientific, Waltham, MA, USA). Sequencing samples were prepared according to the protocol 16S Metagenomic Sequencing Library Preparation for Illumina Miseq System (San Diego, CA, USA). The V3–V4 regions of 16S rDNA gene were amplified. After a purification step with Agencourt AMPure XP (Beckman Coulter Inc., Woerden, The Netherlands), the amplicons were indexed with 10 subsequent cycles of PCR using the Nextera XT Index Kit (Illumina, San Diego, CA, USA). Library sizes were assessed using an Agilent High Sensitivity 2200 Tape Station System (Agilent Technologies, Santa Clara, CA, USA) and quantified by Qubit (ThermoFisher Scientific, Waltham, MA, USA). Normalized libraries were pooled, denatured with NaOH, then diluted to 10pM, and combined with 25% (*v*/*v*) denatured 10pM PhiX, according to Illumina guidelines. A sequencing run was performed on an Illumina Miseq system using v3 reagents for 2 × 301 cycles [40].

### 4.8. Sequences Analysis

The paired-end demultiplexed Illumina sequencing reads were imported into the Quantitative Insights Into Microbial Ecology 2 (QIIME 2; 2021.2 distribution) software suite for downstream analysis. Sequences were then quality-filtered, dereplicated, chimeras identified, and paired-end reads merged in QIIME2 using DADA2; quality filtering was performed using default settings, trimming was set at position 60 (forward and reverse), and truncation lengths were set at 240 for forward and reverse, respectively. A phylogenetic tree was generated using the align-to-tree-mafft-fast tree pipeline in the q2-phylogeny plugin. The Bray–Curtis dissimilarity between samples was calculated using core-metrics-phylogenetic method from the q2-diversity plugin. Classification of Amplicon Sequence Variants (ASVs) was performed using a Bayes algorithm trained using sequences representing the bacterial V3–V4 rRNA region available from the SILVA database2 (Silva_132-99), and the corresponding taxonomic classifications were obtained using the q2-feature-classifier plugin in QIIME2. The classifier was then employed to assign taxonomic information to representative sequences of each ASV. Statistical analyses were performed using the Microbiome analyst software.3 [40].

### 4.9. Bioinformatics Analysis

Quality assessment of sequencing reads will be performed with the prinseq-lite program applying the following parameters: a minimal length (min_length) of 50 nt and a quality score threshold of 30 from the 3′-end (trim_qual_right), using a mean quality score (trim_qual_type) calculated with a sliding window of 20 nucleotides (trim_qual_window). After filtering and trimming, sequences will be analyzed using the qiime2 platform. Sequence de-noising, paired-ends joining, and chimera depletion will be performed with the DADA2 software (version 1.26.0). The taxonomic affiliations of the sequences will be assigned by means of the naïve Bayesian classifier integrated in quiime2 using the SILVA_release_132 database.

### 4.10. Gut Microbiota Analysis and Comparisons

Gut microbiota (GM) composition was analyzed in all experimental groups and compared in order to identify differences and similarities among groups, also according to other clinical characteristics which may have had an impact such as drug therapy and inflammatory state. Microbiota differences were evaluated by comparing α- and β-diversity metrics (QIIME 2; 2021.2 distribution); both were demonstrated by Principal Coordinate Analysis (PCoA). The 97% similarity between sequences was categorized as the same Operational Taxonomic Unit (OTU). To identify specific microbiota signatures, Linear Discriminant Analysis (LDA) effect size (LEfSe) was performed by Microbiome analyst software.3 [40]. LEfSe was set with a *p*-value cutoff of 0.05 and Log LDA score cutoff of 2.0.

### 4.11. Statistical Analysis

We reported quantitative variables as median and interquartile ranges (IQR) and nominal variables as percentages and absolute numbers. Comparisons of continuous variables were performed by the Anova test or the Kruskal–Wallis Test, after testing them for normality using the Shapiro–Wilk test. Differences between categorical variables were assessed by the chi-square test. Differences in the microbial composition between groups were tested using the PERMANOVA test. A *p*-value < 0.05 was considered statistically significant. The data were analyzed using SPSS 26.0 software (IBM Corp, Armonk, NY, USA).

## 5. Conclusions

Our data suggest that in IBD and NAFLD patients, gut inflammation has a greater negative influence on the intestinal microbiota. IBD therefore could drive the alterations that predominantly characterize the intestinal dysbiosis typical of patients also affected by hepatic steatosis. This hypothesis could guide the search for a possible microbial biomarker useful to dive into the pathogenesis of IBD and the pathophysiological networks underlying their complex extraintestinal manifestations and the effectiveness of existing therapies.

Information obtained from metagenomics studies, however, can only allow assumptions about the role of certain species, allowing us to focus on those of major interest based on the existing literature. These findings need to be integrated with novel metabolomics and metaproteomics longitudinal studies that investigate the host–pathogen relationship in these specific diseases, in order to better define the role of microbiota in the gut–liver axis.

## Figures and Tables

**Figure 1 ijms-25-05453-f001:**
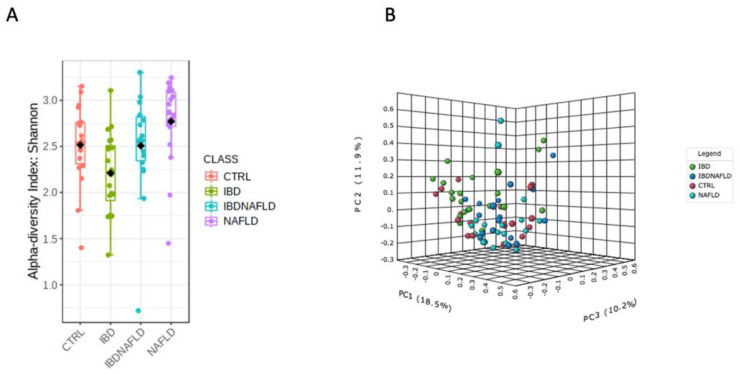
α-diversity assessed by Shannon Index (**A**) and β-diversity assessed by PCoA (**B**) compared between the four study groups. PCoA: Principal Coordinate Analysis.

**Figure 2 ijms-25-05453-f002:**
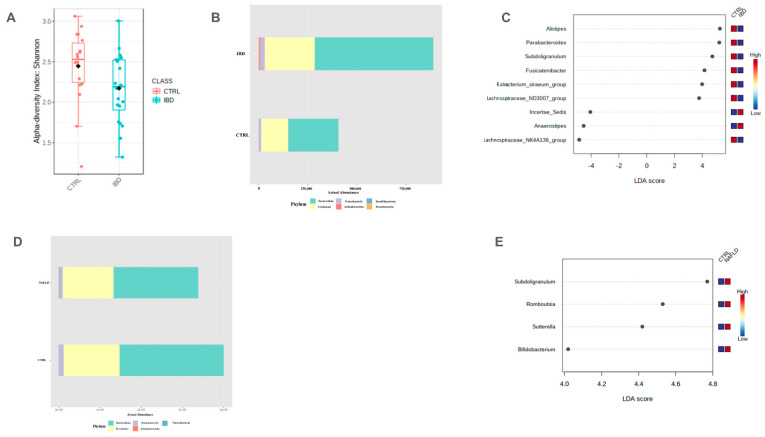
Results in the comparison between IBD-only group and CTRLs according to α-diversity assessed by Shannon Index (**A**), Bacteroidetes/Firmicutes ratio (**B**), and OTUs analysis assessed by LDA score (**C**); between NAFLD-only group and CTRLs according to Bacteroidetes/Firmicutes ratio (**D**), and OTUs analysis assessed by LDA score (**E**). LDA scores can be interpreted as the degree of difference in the relative abundance of OTUs. IBD: Inflammatory Bowel Disease, CTRLs: healthy controls, OTUs: Operational Taxonomic Units, LDA: Linear Discriminant Analysis, NAFLD: Non-Alcoholic Fatty Liver Disease.

**Figure 3 ijms-25-05453-f003:**
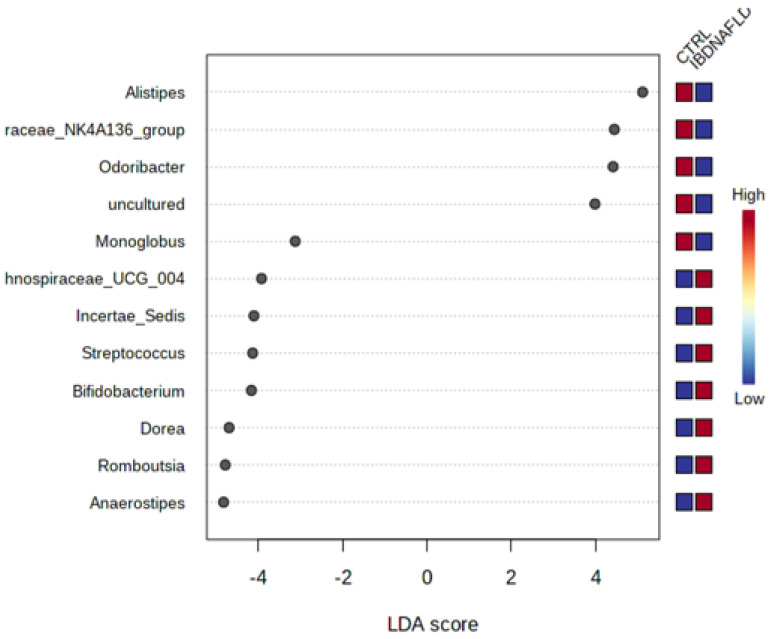
Differential abundance at genus level according to LDA score in the IBD-NAFLD group compared with CTRLs. LDA scores can be interpreted as the degree of difference in the relative abundance of OTUs. LDA: Linear Discriminant Analysis, IBD-NAFLD: Inflammatory Bowel Disease Non-Alcoholic Fatty Liver Disease, CTRLs: healthy controls.

**Figure 4 ijms-25-05453-f004:**
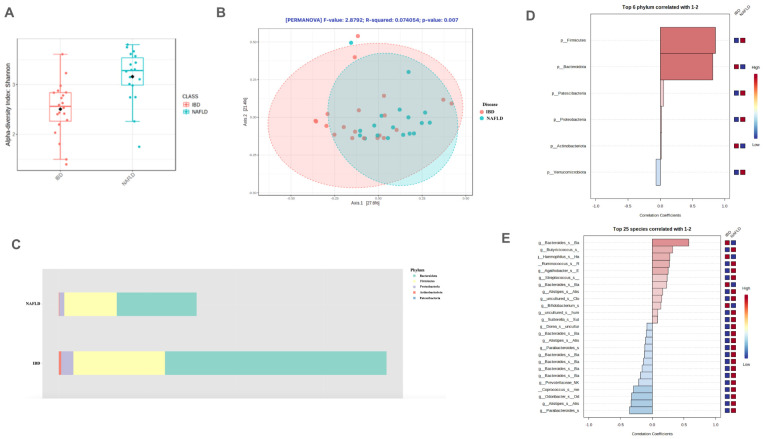
Results in the comparison between IBD-only group and NAFLD-only group according to α-diversity assessed by Shannon Index (**A**), β-diversity assessed by PCoA (**B**), Bacteroidetes/Firmicutes ratio (**C**), differential abundance at phylum (**D**) and species (**E**) level. IBD: Inflammatory Bowel Disease, NAFLD: Non-Alcoholic Fatty Liver Disease, PCoA: Principal Coordinate Analysis.

**Figure 5 ijms-25-05453-f005:**
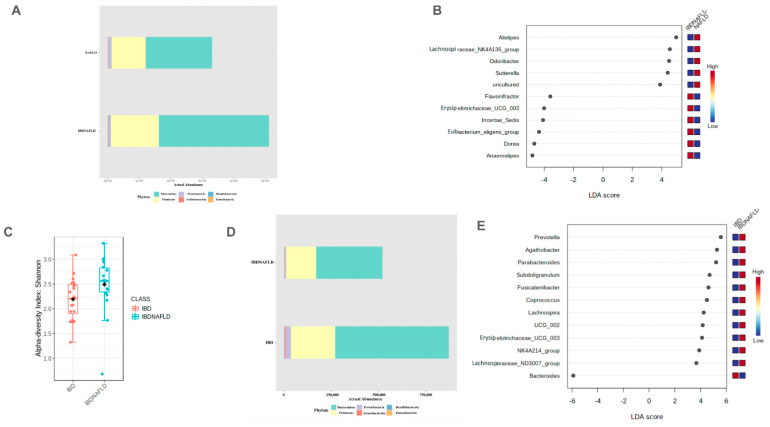
Results in the comparison between NAFLD-only group and IBD-NAFLD group according to Bacteroidetes/Firmicutes ratio (**A**) and differential abundance at genus level according to LDA score (**B**); between IBD-only group and IBD-NAFLD group according to α-diversity assessed by Shannon Index (**C**), Bacteroidetes/Firmicutes ratio (**D**) and differential abundance at genus level according to LDA score (**E**). LDA scores can be interpreted as the degree of difference in the relative abundance of OTUs. IBD: Inflammatory Bowel Disease, NAFLD: Non-Alcoholic Fatty Liver Disease, PCoA: Principal Coordinate Analysis, LDA: Linear Discriminant Analysis.

**Table 1 ijms-25-05453-t001:** Anthropometric and clinical characteristics of the study groups.

	Group 1 IBD-NAFLD	Group 2 IBD-Only	Group 3 NAFLD-Only	Group 4 CTRLs	*p*
N	18	20	18	18	
Age (years)	45 (37, 53)	39 (24, 45)	42 (36, 56)	33 (23, 49)	0.03 ^a^
Gender male *n* (%)	14 (78)	11 (55)	13 (72)	11 (61)	0.44
BMI (Kg/m^2^)	28 (25, 33)	24 (21, 27)	28 (26, 30)	23 (21, 25)	0.001 ^b^
median CAP (dB/m)	280 (263, 323)	202 (179, 220)	284 (255, 316)	206 (185, 237)	0.001 ^c^
median Stiffness (kPa)	5 (4, 6)	5.3 (4, 5)	4.7 (3, 5)	4.4 (4, 6)	0.471
Hb (g/dL)	13.3 (12, 14)	13.2 (11, 14)	13.6 (12, 14)	14.2 (13, 15)	0.04 ^d^
WBC (×10^3^/μL)	6.4 (5, 7)	7.6 (5, 9)	8.5 (6, 9)	9.1 (7, 10)	0.002 ^e^
PLTs (×10^3^/μL)	222 (208, 265)	207 (179, 259)	203 (183, 244)	239 (171, 288)	0.550
CRP (mg/L)	3 (3, 4)	4.3 (3, 8)	3 (3, 4)	3 (3, 6)	0.215
ESR (mm/h)	6.5 (4, 15)	9 (5, 15)	15.5 (3, 22)	8 (4, 13)	0.685
AST (UI/L)	24.5 (17, 49)	16.5 (13, 27)	26.5 (19, 37)	17.5 (15, 22)	0.005 ^f^
ALT (UI/L)	32 (23, 46)	15.5 (10, 19)	33 (24, 44)	15.5 (11, 20)	0.001 ^g^
GGT (UI/L)	32 (14, 37)	17 (10, 34)	25.5 (21, 35)	14.5 (9, 23)	0.002 ^h^
ALP (UI/L)	71.5 (60, 87)	70 (62, 87)	67 (62, 81)	63 (49, 72)	0.211
Tot. Bilirubine (mg/dL)	0.5 (0.4, 0.9)	0.4 (0.2, 0.7)	0.4 (0.3, 0.9)	0.6 (0.3, 0.8)	0.138
Dir. Bilirubine (mg/dL)	0.2 (0.1, 0.2)	0.1 (0.1, 0.2)	0.1 (0.1, 0.2)	0.2 (0.1, 0.3)	0.160
Tot. Cholesterol (mg/dL)	185 (152, 204)	165 (143, 182)	199 (181, 207)	193 (147, 205)	0.003 ^i^
HDL (mg/dL)	46 (39, 52)	46 (42, 51)	42 (36, 48)	51 (43, 63)	0.04 ^l^
LDL (mg/dL)	110 (93, 145)	101 (72, 116)	129 (117, 143)	121 (80, 136)	0.001 ^m^
Triglycerides (mg/dL)	87 (69, 94)	81 (56, 97)	113 (83, 135)	78 (53, 117)	0.107
Glucose (mg/dL)	90 (83, 96)	81 (76, 87)	88 (81, 93)	81 (73, 85)	0.004 ^n^
Iron (mcg/dL)	75 (57, 91)	80 (38, 91)	86 (54, 112)	93 (77, 113)	0.685

Continuous variables are shown as median (interquartile range). Categorical variables are presented as number and proportion. *p* values were calculated by Kruskal–Wallis test. ^a^: statistically significant between groups 2-3, 2-1, 4-3, 4-1. ^b^: statistically significant between groups 4-3, 4-1, 2-3, 2-1. ^c^: statistically significant between groups 2-3, 2-1, 4-3, 4-1. ^d^: statistically significant between groups 2-4, 1-4. ^e^: statistically significant between groups 1-2, 1-3, 1-4. ^f^: statistically significant between groups 2-1, 2-3, 4-1, 4-3. ^g^: statistically significant between groups 4-1, 4-3, 2-1, 2-3. ^h^: statistically significant between groups 4-3, 4-1, 2-1. ^i^: statistically significant between groups 2-1, 2-4, 2-3. ^l^: statistically significant between groups 3-4. ^m^: statistically significant between groups 2-1, 2-4, 2-3, 1-3. ^n^: statistically significant between groups 4-3, 4-1, 2-3, 2-1. Abbreviations: Body Mass Index (BMI), Hemoglobin (Hb), White Blood Cells (WBC), Platelets (PLTs), C-reactive Protein (CRP), Erythrocyte Sedimentation Rate (ESR), Alanine aminotransferase (ALT), Aspartate aminotransferase (AST), Gamma glutamyl transferase (GGT), Alkaline phosphatase (ALP), High density lipoprotein cholesterol (HDL), Low density lipoprotein cholesterol (LDL).

**Table 2 ijms-25-05453-t002:** Anthropometric and clinical characteristics of IBD patients stratified by the presence or absence of NAFLD.

	Group 1 IBD-NAFLD	Group 2 IBD-Only	*p*
N	18	20	
Age (years)	45 (37, 53)	39 (24, 45)	0.03
Gender *n* male (%)	14 (78)	11 (55)	0.2
BMI (Kg/m^2^)	28 (25, 33)	24 (21, 27)	0.003
median CAP (dB/m)	280 (263, 323)	202 (179, 220)	0.000
median Stiffness (kPa)	5 (4, 6)	5.3 (4, 5)	0.965
Disease duration (years)	13 (5, 17)	5 (2, 11)	0.017
MAYO Score	0.5 (0, 1)	1 (0, 2)	0.497
Harvey–Bradshaw	6.5 (3, 7)	4.5 (0, 9)	0.9
Disease activity, years *n* (%)	1 (5)	3 (15)	0.6
CRP (mg/L)	3 (3, 4)	4.3 (3, 8)	0.067
ESR (mm/h)	6.5 (4, 15)	9 (5, 15)	0.426
Calprotectin (mcg/g)	36 (22, 99)	82 (16, 383)	0.264
Disease Characteristics			
IBD to use *n* (%)			0.34
Crohn’s Disease *n* (%)	6 (33)	10 (50)	
Ulcerative Colitis *n* (%)	12 (67)	10 (50)	
Disease extension (CD) *n* (%)			0.60
Ileitis *n* (%)	2 (11)	2 (10)	
Ileo-colitis *n* (%)	4 (22)	8 (40)	
Disease extension (UC)			0.86
Proctitis *n* (%)	1 (5)	1 (5)	
Left colitis *n* (%)	1 (5)	2 (10)	
Pancolitis *n* (%)	10 (55)	7 (35)	
Treatment			
Mesalamine y *n* (%)	15 (83)	15 (75)	0.7
Steroids y *n* (%)	0	3 (15)	0.23
Azathioprine y *n* (%)	0	0	
Biologics y *n* (%)	7 (39)	11 (55)	0.35

Continuous variables are shown as median (interquartile range). Categorical variables are presented as number and proportion. *p* values were calculated by Mann–Whitney or by chi-square test for continuous or categorical traits, respectively.

**Table 3 ijms-25-05453-t003:** Periodontal health scores.

Score	Gingival Index, GI	Plaque Index, PI
0	normal gingiva	absence of microbial plaque
1	mild inflammation (i.e., slight change in color, slight edema, no bleeding on probing)	thin film of microbial plaque along the free gingival margin
2	moderate inflammation (i.e., redness, edema, and glazing, or bleeding on probing)	moderate accumulation with plaque in the sulcus
3	severe inflammation (i.e., marked redness and edema, tendency toward spontaneous bleeding, ulceration)	large amount of plaque in sulcus or pocket along the free gingiva margin

## Data Availability

The raw data supporting the conclusions of this article will be made available by the authors on request.

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
