# Peer review of "Gut Microbiota Profile Changes in Patients with Inflammatory Bowel Disease and Non-Alcoholic Fatty Liver Disease: A Metagenomic Study"

_ijms, 2024, doi:10.3390/ijms25105453_

Round 1

Reviewer 1 Report

Comments and Suggestions for Authors

The significance of this study is not clear.

The results seems insignificant and the practical relevance of this study seems low.

Further proteome and metabolome based analysis should be conducted and the effect of metabolites produced in different diseases and their effect in mediating the diseases should be emphasized.

The representation of figures is low and should be improved

Comments on the Quality of English Language

The quality of language used should be improved and also there are numerous grammatical and typographical errors in the manuscript.

Author Response

REVIEWER 1

The significance of this study is not clear.The results seems insignificant and the practical relevance of this study seems low.

Further proteome and metabolome based analysis should be conducted and the effect of metabolites produced in different diseases and their effect in mediating the diseases should be emphasized.

The representation of figures is low and should be improved

Thank you for your feedback. The study examines taxonomic changes in the gut microbiota in different cohorts of patients with IBD and NAFLD, alone or in combination.

We agree with the reviewer's point of view and realize that not having performed a functional assessment of the microbiota through metabolomics and metaproteomics techniques is a limitation. The importance of this additional information, however, was mentioned in the discussion and conclusions.

Metagenomics studies do not consider the dynamic host-pathogen relationship in the disease context, however, they represent a valuable tool for analyzing the extent of dysbiosis.

The significance of our study consists in having considered a group of patients with both diseases.

The size of the figures has been improved.

Reviewer 2 Report

Comments and Suggestions for Authors

The study presents valuable insights into the gut microbiota profiles of patients with IBD and NAFLD, shedding light on the intricate relationship between gut inflammation and microbial composition. However, several limitations need to be addressed to enhance the robustness of the findings.

1. The 16S rDNA gene is present in the genome of all bacteria and is highly conserved. The sequence contains 9 highly variable regions and 10 conserved regions, and a sequence of about 1500 bp is obtained by PCR amplification of a particular segment of highly variable region sequence (V4 region or V3-V4 region) followed by sequencing. Metagenomics sequencing, on the other hand, involves randomly interrupting the microbial genomic DNA into small fragments of 500 bp, and then adding universal primers to the ends of the fragments for PCR amplification and sequencing, and then splicing the small fragments into longer sequences by means of assembly.

As can be seen from the Materials and Methods section of this paper, 16S rDNA sequencing, rather than metagenomics sequencing, was performed in this study. However, this article mentions several times that this study performed metagenomics analysis (including the title of the article). This is very inappropriate and makes readers wonder whether the authors are clear about the difference between 16S rDNA sequencing and metagenomics sequencing.

2. Line 91, CAP. For abbreviations that appear for the first time in the text, the full name should be indicated. Please check for similar problems and revise.

3. The resolution of the figures in the article is so low that the words in the legend are not recognizable even when enlarged.

4. Why are the criteria not uniform when screening for differential taxonomies using LDA values? Line 176, LEfSe: LDA score > 3.0, Line 185, LEfSe: LDA score > 4.0, etc.

5. How were the PCoA and LEfSe analyses done? Please give a more detailed description, i.e., what software or platform was used.

6. The discussion section has too much repetition of results and lacks real in-depth discussion.

Comments on the Quality of English Language

Minor editing of English language required.

Author Response

REVIEWER 2

  1. The 16S rDNA gene is present in the genome of all bacteria and is highly conserved. The sequence contains 9 highly variable regions and 10 conserved regions, and a sequence of about 1500 bp is obtained by PCR amplification of a particular segment of highly variable region sequence (V4 region or V3-V4 region) followed by sequencing. Metagenomics sequencing, on the other hand, involves randomly interrupting the microbial genomic DNA into small fragments of 500 bp, and then adding universal primers to the ends of the fragments for PCR amplification and sequencing, and then splicing the small fragments into longer sequences by means of assembly.

As can be seen from the Materials and Methods section of this paper, 16S rDNA sequencing, rather than metagenomics sequencing, was performed in this study. However, this article mentions several times that this study performed metagenomics analysis (including the title of the article). This is very inappropriate and makes readers wonder whether the authors are clear about the difference between 16S rDNA sequencing and metagenomics sequencing.

Thank you for your comment. In the Material and Methods section, we reported the employed protocol to perform the taxonomic classification of microbiota: the 16S Metagenomic Sequencing Library Preparation by Illumina. According to the protocol of the producer, Metagenomic studies are commonly performed by analyzing the prokaryotic 16S ribosomal RNA gene (16S rRNA), which is approximately 1,500 bp long and contains nine variable regions interspersed between conserved regions. Variable regions of 16S rRNA are frequently used in phylogenetic classifications such as genus or species in diverse microbial populations. The protocol describes a method for preparing samples for sequencing the variable V3 and V4 regions of the 16S rRNA gene.

Based on our experience, metagenomics sequencing includes several analyses, such as 16S sequencing to determine microbiota composition. In this view, we specified the analysis in the title and through the text.

  1. Line 91, CAP. For abbreviations that appear for the first time in the text, the full name should be indicated. Please check for similar problems and revise.

We proceeded to specify the meaning of all acronyms.

  1. The resolution of the figures in the article is so low that the words in the legend are not recognizable even when enlarged.

We improved the resolution of all figures.

  1. Why are the criteria not uniform when screening for differential taxonomies using LDA values? Line 176, LEfSe: LDA score > 3.0, Line 185, LEfSe: LDA score > 4.0, etc.

Thank you for your comment. In the Material and Methods section, we specified the threshold value, adding the following sentence: “LEfSe was set with p-value cutoff 0.05 and Log LDA score cutoff 2.0.”. Legend of Figure 4 was modified deleting “according to LDA score.” and the spelling of acronym. Subsequently, all results were reported considering LEfSe: LDA score > 2. Additionally, in the results we highlighted the negative LDA score, LDA scores can be interpreted as the degree of difference in the relative abundance of OTUs.

  1. How were the PCoA and LEfSe analyses done? Please give a more detailed description, i.e., what software or platform was used.

Thank you for your comment. In the final part of 4.10 Gut microbiota analysis and comparisons paragraph (Material and Methods section) we modified the last sentence as following: “To identify specific microbiota signatures, Linear discriminant analysis (LDA) effect size (LEfSe) was performed by Microbiome analyst software.3 [39].”

  1. The discussion section has too much repetition of results and lacks real in-depth discussion.

The discussion has been edited by removing all repetitions regarding information already stated in the results. We have expanded the discussion by citing additional work that may corroborate the assumptions made on the basis of our results. References to future perspectives regarding the role of the microbiota in the context of IBD and MAFLD have been added.

Reviewer 3 Report

Comments and Suggestions for Authors

The article " Gut microbiota profile changes in patients with Inflammatory Bowel Disease and Non-Alcoholic Fatty Liver Disease: a metagenomic study" presents an interesting literature topic. However, the article requires improvements:

1.               The introduction section should be expanded. Please provide more information regarding the microbiota.

2.               The Materials and Methods section should be placed after the Introduction.

3.               The Discussion section requires improvement. It would greatly help to incorporate information regarding treatment - specifically, the potential benefits of microbiota transfer in liver cirrhosis and NAFLD - for example: https://doi.org/10.3390/biomedicines11112930, but there are other sources to draw from as well. Additionally, you could discuss microRNAs in digestive pathology and beyond. For example: https://doi.org/10.3390/biomedicines11072058. Today, microRNAs provide valuable insights, both through their interaction with specific genes and through the identification of patterns depending on the underlying pathology.

4.               The conclusions need to better incorporate the information from the study.

Comments on the Quality of English Language

 Minor editing of English language required.

Author Response

REVIEWER 3

  1. The introduction section should be expanded. Please provide more information regarding the microbiota.

Thank you for your comment. The introduction has been expanded by adding references to the most recent work regarding the gut microbiota and its connection to IBD and NAFLD. We added some reference to therapeutic perspectives as well, citing two studies on the use of probiotics and FMT in these diseases.

  1. The Materials and Methods section should be placed after the Introduction.

The layout of the manuscript complies with the journal guidelines, which suggest that the “materials and methods” section should be included at the end.

  1. The Discussion section requires improvement. It would greatly help to incorporate information regarding treatment - specifically, the potential benefits of microbiota transfer in liver cirrhosis and NAFLD - for example: https://doi.org/10.3390/biomedicines11112930, but there are other sources to draw from as well. Additionally, you could discuss microRNAs in digestive pathology and beyond. For example: https://doi.org/10.3390/biomedicines11072058. Today, microRNAs provide valuable insights, both through their interaction with specific genes and through the identification of patterns depending on the underlying pathology.
  2. The conclusions need to better incorporate the information from the study.

We have expanded the discussion by citing additional work that may corroborate the assumptions made on the basis of our results. References to future perspectives regarding the role of the microbiota in the context of IBD and MAFLD have been added, also citing the study on the pathogenetic implications of miRNA suggested by the reviewer. We made sure that the conclusions of the study were more detailed, focusing on its actual outcome.

Reviewer 4 Report

Comments and Suggestions for Authors

De Caro et al. report in this paper (ijms-2934661), with title “Gut microbiota profile changes in patients with Inflammatory Bowel Disease and Non-Alcoholic Fatty Liver Disease: a metagenomic study”, on the study on the composition of the gut microbiota of Inflammatory Bowel Disease (IBD) and Non-Alcoholic Fatty Liver Disease (NAFLD) patients. The authors compare their results with each disorder considered separately and with a control group (CTRLs), claiming that in these patients the gut inflammation has a greater negative influence on the intestinal microbiota. Overall, the methodology and research work are well performed and reach the standard of the journal. Here are some important points to be revised:

1) Some abbreviations (CAP, BMI, NASH, PPI and so on ) must be defined in the main text before writing their abbreviation.

2) The size of some Figures (Figures 2B, 2C, 2D, 4B, 4C, 4D, 4E, 5A, 5D and 5E) should be increased, given that there are some words (in their legends) which are very difficult to read.

3) Is there any idea raised from the work of why the LDL and Cholesterol levels are lower in the group 2 (IBD-only)?, as indicated in Table 1.

4) More details about the method of evaluation of calprotectin and its immunochemical technique should be included, also indicating the reference ranges of this important 'bio-marker'.

5) The conclusions section should be further developed, it is too short as it is and should be expanded. Please, check this out. 

Comments on the Quality of English Language

Minor editing of English language is required.

Author Response

REVIEWER 4

1) Some abbreviations (CAP, BMI, NASH, PPI and so on ) must be defined in the main text before writing their abbreviation.

2) The size of some Figures (Figures 2B, 2C, 2D, 4B, 4C, 4D, 4E, 5A, 5D and 5E) should be increased, given that there are some words (in their legends) which are very difficult to read.

Thank you for your comments. We proceeded to specify the meaning of all acronyms. The size of the figures has also been improved.

3) Is there any idea raised from the work of why the LDL and Cholesterol levels are lower in the group 2 (IBD-only)?, as indicated in Table 1.

This finding could be explained by the fact that Group 2, unlike Group 1 and Group 3, includes patients with IBD only, in whom hepatic steatosis was ruled out. LDL cholesterol levels, in fact, as known from literature, positively correlate with some dysmetabolic conditions such as NAFLD.

4) More details about the method of evaluation of calprotectin and its immunochemical technique should be included, also indicating the reference ranges of this important 'bio-marker'.

We proceeded to better specify our laboratory's fecal calprotectin assay method and reference ranges.

 5) The conclusions section should be further developed, it is too short as it is and should be expanded. Please, check this out. 

As replied to reviewers 2 and 3, we made sure that the discussion and conclusions of the study were more detailed, focusing on its actual outcome.

Reviewer 5 Report

Comments and Suggestions for Authors

Comments to the authors:

In this study, the authors investigated changes in gut microbiota profiles among patients diagnosed with inflammatory bowel disease (IBD) and non-alcoholic fatty liver disease (NAFLD) using a meta-genomic analysis. They observed significant differences in bacterial α-diversity, with the IBD-only group displaying notably lower values compared to other groups, particularly when contrasted with the IBD-NAFLD group. Additionally, β-diversity exhibited significant variations among the groups. The study revealed a higher Bacteroidetes/Firmicutes ratio exclusively when comparing IBD groups to healthy controls (CTRLs). A decrease in the differential abundance of Subdoligranulum, Parabacteroides, and Fusicatenibacter was noted when comparing the IBD-only group with the IBD-NAFLD group. Conversely, comparing the NAFLD-only group with the IBD-NAFLD group revealed a higher abundance of Alistipes, Odoribacter, Sutterella, and Lachnospira. The comparison between the IBD-only and other groups indicated an inverse relationship. Ultimately, the authors concluded that, for the first time, they demonstrated the distinct fecal microbial composition in IBD patients with concurrent NAFLD, contrasting with the characteristic microbial signatures known for each disorder individually. Overall, this study is interesting, and the authors provide some exciting data. I have several concerns that need to be addressed by the authors before publication. The Major concerns see below:

1: Potential Confounding Factors: The study doesn't adequately address potential confounding factors such as dietary habits, lifestyle factors, and medication history, which could influence gut microbiota composition and disease pathogenesis. Controlling for these variables or providing information on them would strengthen the validity of the results.

2: Lack of Longitudinal Data: Cross-sectional studies like this one provide a snapshot of gut microbiota composition at a single point in time. Longitudinal studies would offer insights into how the microbiota changes over time and its relationship with disease progression.

3: Absence of Functional Analysis: While the study assesses microbial composition, it lacks functional microbiota analysis. Understanding the functional roles of different microbial species or communities could provide deeper insights into their involvement in disease pathogenesis.

4: Interpretation of Results: While the study reports differences in gut microbiota composition among different groups, it's essential to interpret these findings cautiously. Correlation does not imply causation, and the observed microbial differences may be a consequence rather than a cause of the diseases studied.

5: Statistical Considerations: The statistical methods employed in the study should be carefully evaluated to ensure appropriate handling of data and interpretation of results, including adjustments for multiple comparisons and controlling for potential confounders.

Comments on the Quality of English Language

Minor editing of English language required

Author Response

REVIEWER 5

1: Potential Confounding Factors: The study doesn't adequately address potential confounding factors such as dietary habits, lifestyle factors, and medication history, which could influence gut microbiota composition and disease pathogenesis. Controlling for these variables or providing information on them would strengthen the validity of the results.

2: Lack of Longitudinal Data: Cross-sectional studies like this one provide a snapshot of gut microbiota composition at a single point in time. Longitudinal studies would offer insights into how the microbiota changes over time and its relationship with disease progression.

3: Absence of Functional Analysis: While the study assesses microbial composition, it lacks functional microbiota analysis. Understanding the functional roles of different microbial species or communities could provide deeper insights into their involvement in disease pathogenesis.

4: Interpretation of Results: While the study reports differences in gut microbiota composition among different groups, it's essential to interpret these findings cautiously. Correlation does not imply causation, and the observed microbial differences may be a consequence rather than a cause of the diseases studied.

Thank you for your feedback.

As specified in the materials and methods and discussion sections, some potential confounding factors were addressed using valuable strategies. For example, the patients enrolled in the study were also matched by dietary habits using the Food Frequency Questionnaire. Regarding treatments, we conducted a subanalysis to verify that different therapies did not affect outcomes in a statistically significant way (data not shown). Our sample size, as already specified, was small, and most of the patients underwent the same type of treatment.

As replied to reviewer 1, the study examines only taxonomic changes in the gut microbiota of the patients enrolled.

Functional assessment of the microbiota, conducted through metabolomics and metaproteomics techniques, is beyond the scope of the present study. The importance of this additional information, however, was mentioned in the discussion and conclusions, along with the practical utility of a longitudinal design.

Metagenomics studies do not consider the dynamic host-pathogen relationship in the disease context, however, they represent a valuable tool for analyzing the extent of dysbiosis. Finally, the significance of our study consists in having considered a group of patients with both diseases.

 5: Statistical Considerations: The statistical methods employed in the study should be carefully evaluated to ensure appropriate handling of data and interpretation of results, including adjustments for multiple comparisons and controlling for potential confounders.

We thank the reviewer for the comment. As described in the section on statistical methodology, we conducted statistical analyses using multiple comparisons between the study groups.

Reviewer 6 Report

Comments and Suggestions for Authors

In the study entitled Gut microbiota profile changes in patients with Inflammatory Bowel Disease and Non-Alcoholic Fatty Liver Disease: a metagenomic study the authors to provide compared gut microbial composition in patients diagnosed with either IBD or NAFLD or a combination of both.

With this very interesting work, the authors delve deeper into the theme of the review https://doi.org/10.3390/ijms25063278 which I suggest including in the bibliography.

My suggestions and recommendations:

The authors write in results: The participants were stratified into four groups according to IBD diagnosis.... The first group, named IBD-NAFLD, included 18 IBD patients, 78% (n=14) were male....The second group, named IBD-only group, included 20 IBD patients, 55% (n=11) were male.....

The data regarding women is not clear in the tables and in the description of the results. I therefore ask you to organize the data highlighting the differences between men and women. The wider attention to the gender (sex) different would represent an added value also for current knowledge regarding the influence of gender (sex) on the gut microbiota and pathologies.

I think that, might be made better by updating the bibliography and it would be better to fix the "plagiarism" lines.

With a few small changes, the text can be accepted as is.

Author Response

The authors write in results: The participants were stratified into four groups according to IBD diagnosis.... The first group, named IBD-NAFLD, included 18 IBD patients, 78% (n=14) were male...The second group, named IBD-only group, included 20 IBD patients, 55% (n=11) were male.....The data regarding women is not clear in the tables and in the description of the results. I therefore ask you to organize the data highlighting the differences between men and women. The wider attention to the gender (sex) different would represent an added value also for current knowledge regarding the influence of gender (sex) on the gut microbiota and pathologies.

We thank the reviewer for the suggestion. We have included in Table 1 the percentage of male gender distributed among the four groups, which, as observed, do not show significant differences.

I think that, might be made better by updating the bibliography and it would be better to fix the "plagiarism" lines.

We have worked  on revising and updating the bibliography to ensure the completeness and accuracy of the cited sources. Furthermore, we subjectedmanuscript  to the Compilatio software, which indicated a plagiarism level of 6%. If requested, we will provide the generated report."

Round 2

Reviewer 1 Report

Comments and Suggestions for Authors

The authors should add a few lines displaying the novelty, and significance of this study in abstract

A separate paragraph at the end of the introduction revealing the significance and application of this study 

A thorough discussion should be included revealing the association between the finding of this study and the choice of treatments available. 

Also, the conclusion should be modified to reveal how this study can benefit in understanding the pathogenesis of the two diseases and in development of treatment

Comments on the Quality of English Language

There are few typographical and grammatical errors in the manuscript which needs correction.

Author Response

The authors should add a few lines displaying the novelty, and significance of this study in abstract.
A separate paragraph at the end of the introduction revealing the significance and application of this study
A thorough discussion should be included revealing the association between the finding of this study and the choice of treatments available.
Also, the conclusion should be modified to reveal how this study can benefit in understanding the pathogenesis of the two diseases and in development of treatment.

Thank you for the suggestions.
We have added new paragraphs in the introduction, discussion and conclusion to explain the possible applications of our findings in studying both the pathogenesis of IBD and their extraintestinal manifestations. Regarding the choice and efficacy of therapies, although our results were not particularly
directing, we added a few lines to mention the possible role of metagenomics in finding asuitable biomarker, citing a recent meta-analysis that investigated this important aspect ([32] Wang C, Gu Y, Chu Q et al. Gut microbiota and metabolites as predictors of biologics response in inflammatory bowel disease: A comprehensive systematic review. MicrobiolRes. 2024 May;282:127660).

Reviewer 3 Report

Comments and Suggestions for Authors

The authors have made the requested changes.

Author Response

We thank you for the response."

Round 3

Reviewer 1 Report

Comments and Suggestions for Authors

The manuscript may be accepted after thorough English language check

Comments on the Quality of English Language

The manuscript may be accepted after thorough English language check